# Unexpected differences in the pharmacokinetics of N-acetyl-DL-leucine enantiomers after oral dosing and their clinical relevance

Grant C. Churchill[1]*, Michael Strupp[2], Antony Galione[1], Frances M. Platt[1]

**1** Department of Pharmacology, University of Oxford, Oxford, United Kingdom, **2** Department of Neurology, German Center for Vertigo and Balance Disorders, Ludwig Maximilians University Hospital Munich, Munich, Germany

* grant.churchill@pharm.ox.ac.uk

**Data Availability Statement:** All relevant data are within the paper and its Supporting Information files.

## Abstract

The enantiomers of many chiral drugs not only exhibit different pharmacological effects in regard to targets that dictate therapeutic and toxic effects, but are also handled differently in the body due to pharmacokinetic effects. We investigated the pharmacokinetics of the enantiomers of N-acetyl-leucine after administration of the racemate (N-acetyl-DL-leucine) or purified, pharmacologically active L-enantiomer (N-acetyl-L-leucine). The results suggest that during chronic administration of the racemate, the D-enantiomer would accumulate, which could have negative effects. Compounds were administered orally to mice. Plasma and tissue samples were collected at predetermined time points (0.25 to 8 h), quantified with liquid chromatography/mass spectrometry, and pharmacokinetic constants were calculated using a noncompartmental model. When administered as the racemate, both the maximum plasma concentration ($C_{max}$) and the area under the plasma drug concentration over time curve (AUC) were much greater for the D-enantiomer relative to the L-enantiomer. When administered as the L-enantiomer, the dose proportionality was greater than unity compared to the racemate, suggesting saturable processes affecting uptake and/or metabolism. Elimination ($k_e$ and $T_{1/2}$) was similar for both enantiomers. These results are most readily explained by inhibition of uptake at an intestinal carrier of the L-enantiomer by the D-enantiomer, and by first-pass metabolism of the L-, but not D-enantiomer, likely by deacetylation. In brain and muscle, N-acetyl-L-leucine levels were lower than N-acetyl-D-leucine, consistent with rapid conversion into L-leucine and utilization by normal leucine metabolism. In summary, the enantiomers of N-acetyl-leucine exhibit large, unexpected differences in pharmacokinetics due to both unique handling and/or inhibition of uptake and metabolism of the L-enantiomer by the D-enantiomer. Taken together, these results have clinical implications supporting the use of N-acetyl-L-leucine instead of the racemate or N-acetyl-D-leucine, and support the research and development of only N-acetyl-L-leucine.

**Funding:** This study was financially supported by IntraBio (https://intrabio.com). The authors (GCC, MS, AG and FMP) were paid for consultancy work for IntraBio. Authors, acting in their capacity as consultants for IntraBio, played roles in study design, data collection and analysis, decision to publish, or preparation of the manuscript. FMP is also a Royal Society Wolfson Research Merit Award holder and a Wellcome Trust Investigator in Science.

**Competing interests:** I have read the journal's policy and the authors of this manuscript have the following competing interests: MS is Joint Chief Editor of the Journal of Neurology, Editor in Chief of Frontiers of Neuro-otology and Section Editor of F1000. He has received speaker's honoraria from Abbott, Actelion, Auris Medical, Biogen, Eisai, Grünenthal, GSK, Henning Pharma, Interacoustics, Merck, MSD, Otometrics, Pierre-Fabre, TEVA, UCB. He is a shareholder of IntraBio. He acts as a consultant for Abbott, Actelion, AurisMedical, Heel, IntraBio and Sensorion. GCC, AG and FMP are cofounders, shareholders and consultants to IntraBio. FMP is a consultant to Actelion. IntraBio Ltd is the applicant for patents WO2018229738 (Treatment For Migraine), WO2017182802 (Acetyl-Leucine Or A Pharmaceutically Acceptable Salt Thereof For Improved Mobility And Cognitive Function), WO2019078915 and WO2018029658 (Therapeutic Agents For Neurodegenerative Diseases), WO2018029657 (Pharmaceutical Compositions And Uses Directed To Lysosomal Storage Disorders), and WO2019079536 (Therapeutic Agents For Improved Mobility And Cognitive Function And For Treating Neurodegenerative Diseases And Lysosomal Storage Disorders). This does not alter our adherence to PLOS ONE policies on sharing data and materials.

## Introduction

N-acetyl-leucine has been used as an over-the-counter drug for the treatment of vertigo since 1957 [1]. Although the mechanism of action of N-acetyl-leucine for vertigo is not known, evidence implicates direct action on the central vestibular-related pathways and ocular motor networks [2,3], as also demonstrated by [18F]fluorodeoxyglucose and Positron Emission Tomography [4]. At the molecular level, several mechanisms of action have been suggested including physicochemical partitioning into the phopspholipid bilayer to decrease its fluidity [1], direct action on glycine recptors and AMPA receptors [1], effects on branched chain aminotransferaes affecting glutamate neurotransmission [3], and an increase in glucose metabolism [3]. All of these mechanisms could contrribute to the observed effect of norrmalizing membrane potential and excitability leading to activation of the vestibulocerebellum and a deactivation of the posterolateral thalamus [2,3,5].

Recently, N-acetyl-leucine has experienced a renaissance with renewed interest from both academia and industry as a promising treatment for several disorders with unmet medical needs including cerebellar ataxia [3,6–8], cognition and mobility in the elderly [9], lysosomal storage disorders [10,11] migraine [12] and restless legs syndrome [13]. Given the broad therapeutic potential of N-acetyl-leucine, its pharmacodynamics and pharmacokinetics warrant further exploration.

As N-acetyl-leucine is an analogue of the alpha amino acid leucine, and because it retains leucine's stereocentre it exists as a pair of enantiomers (Fig 1). Enantiomers are isomers, compounds with the same molecular formula but which differ in the arrangement of their atoms in space, having one chiral stereocentre with four different substituents that yields two non-superimposable mirror image molecules (Fig 1). Often the pharmacological activity of a drug resides with a single enantiomer because living systems are chiral and formed from chiral constituents [14]. Thus, proteins encoded by mRNA and synthesized on ribosomes from L-amino acids are chiral and show stereoselective binding of drugs to transporters, receptors and enzymes [14]. Stereoselective binding can be trivial or profound: S-asparagine is sweet whereas R-asparagine is bitter; R-thalidomide is a sedative whereas the S-form is teratogenic [15]. The thalidomide tragedy shifted the importance of drug chirality from inconsequential to crucial [16] as reflected by the current requirements for developing single enantiomers in drug development and regulatory approval [17,18].

The effects of chirality on drug behaviour has shifted from nice to know to effectively need to know basis for informed dosing and regulatory compliance, safety and efficacy. As N-acetyl-leucine was developed before realization of the importance of drug chirality, it was and continues to be used and marketed as a racemate (Tanganil®, Laboratoires Pierre Fabre)[1]. Subsequent studies in models of vertigo on the individual enantiomers have revealed that the therapeutic effects of N-acetyl-DL-leucine are due to the L-enantiomer [4,19]. This means that, as expressed by Ariens [16] for chiral drugs in general, the racemic mixture (N-acetyl-DL-leucine) is in fact two drugs (the L-enantiomer and the D-enantiomer), each with distinct properties with one (N-acetyl-D-leucine) at best that does not contribute to the therapeutic response, and at worst potentially responsible for toxicity. Indeed, inclusion of an inactive enantiomer provides nothing but an impurity or 'isomeric ballast, as coined by Ariens [16,20]. The resulting concern was stated by Lees et al. [21]: "An impurity at the level 50% of the active constituent would never be tolerated by regulatory authorities for any other constituent, which is neither an active, nor excipient nor solvent.".

Chirality affects not only the pharmacodynamic properties of potency, efficacy and affinity, it also affects pharmacokinetic processes of absorption, distribution, metabolism and excretion [21,22]. Accordingly, both the US Food and Drug Administration's Guidance for Industry on

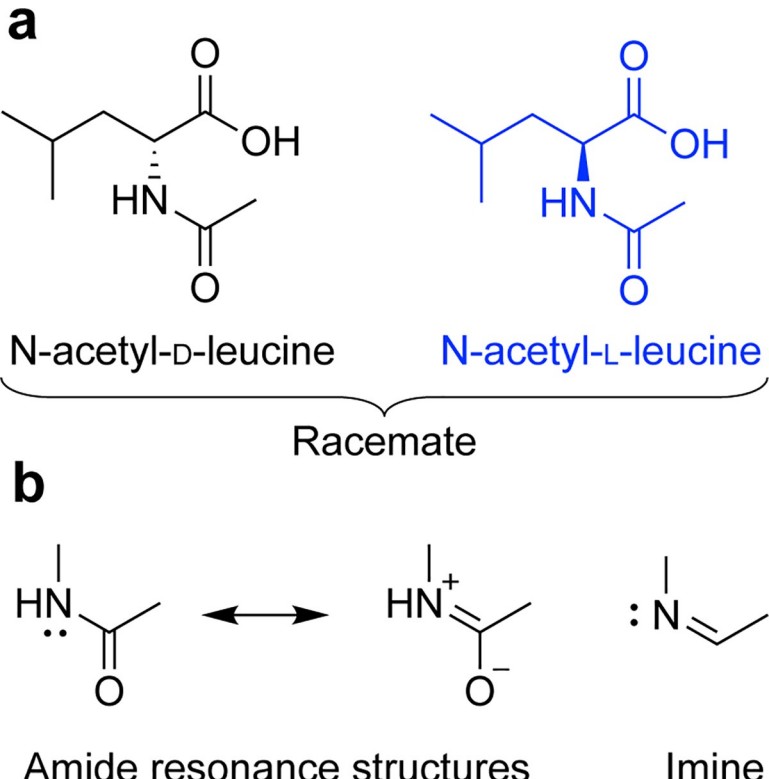

**Fig 1. Chemical structure of N-acetyl-leucine.** (**a**) Stereochemistry of the enantiomers. (**b**) Amide resonance structures showing similarity to an imine. Extending from the tetrahedral chiral carbon is a solid wedge to indicate a bond projecting above the plane of the paper and a hashed wedge to indicate a bond projecting below the plane of the paper.

the Development of New Stereoisomeric Drugs [17] and the European Medicines Agency [18] advises that when a single enantiomer has been found to be the pharmacologically active ingredient of a racemic mixture, it is important to not only characterize the pharmacokinetics of the active enantiomer, but also the effects of the inert member of the stereoisomer pair to determine if there are potential risks with administering the racemate, or benefits associated with the use of the single active enantiomer.

With the above as background regarding safety and efficacy of racemic drugs, and the fact that no data has been published on the pharmacokinetics of the enantiomers of N-acetyl-leucine, we investigated the pharmacokinetics of the racemate (an equal mixture of D and L) as well as the pharmacologically active L-enantiomer alone. We report significant and unexpected differences in the pharmacokinetics of the enantiomers.

## Materials and methods

### Animal ethics approval

All experimental work in this study was conducted by Admescope Ltd. (Oulu, Finland) and was prospectively approved by the national Animal Experiment Board of Finland. The study was conducted under the project licence ESAVI/3047/04.10.07/2016, approved by the national Animal Experiment Board of Finland under Directive 2010/63/EU of the European Parliament and of the Council of 22 September 2010 on the protection of animals used for scientific

purposes, with the following national provisions: Act (497/2013) and Decree (564/2013) on the Protection of Animals Used for Scientific or Educational Purposes (564/2013).

The Animal Welfare Body (AWB) of Admescope Ltd. oversees the animal care and use program. The activity of AWB is documented and it is monitored by the Provincial Veterinarian Officer. The Admescope laboratory Animal Unit is authorized, monitored and regularly inspected by the Regional State Administrative Agency of Southern Finland (Provincial veterinarian officer). The National Animal Experiment Board in Finland is responsible for the protocol review and authorisation. Approval number: ESAVI/3047/04.10.07/2016.

## Details of animal welfare

Animals (5–8 weeks of age during the experiment) were purchased from Scanbur (Denmark) from Charles River Laboratories (Germany) and were housed in individually ventilated-cages in groups of six mice. The cages were provided with aspen bedding (4HP and PM90L, Tapvei, Estonia) and paper strands (Sizzlenest, Datesand, UK) as nesting material, and a paper pulp cabin and red polycarbonate cylinder (Datesand, UK) as cage enrichment. The temperature ($22 \pm 2°C$), humidity ($55 \pm 10\%$) and air exchange rate (75 (times/h) of the individually ventilated-cages and 12/12-h light/dark cycle (500 lux lighting on at 6 am, 1.5 lux lighting on at 6 pm) of the animal holding room were automatically controlled and maintained. Animals were allowed to acclimatize to the site for at least five days prior to the study. Animals had *ad libitum* access to food (SDS diets, RM1 (E) 801002, Special Diets Services, UK) and tap water at all times, and their welfare was assured with daily observations.

To minimize animal suffering and distress, clinical signs and general behaviour of the animals were recorded when necessary. Noninvasively monitoring enables detection of any signs that a drug was not well tolerated in this species and strain. Note that no adverse effects were noted. The animals were not anesthetized during administration of study formulations or during blood sampling, as the pain inflicted by these procedures is considered as minor. Additionally, the following internationally acknowledged primary standards, regulations, recommendations and legislation are applied to the institutional animal care and use program to support animal welfare, acknowledging 3R principles and transparency at Admescope Ltd: 1, European Convention for the Protection of Vertebrate Animals Used for Experimental and Other Scientific Purposes, Council of Europe (ETS 123); 2, The Guide for the Care and Use of Laboratory Animals, NRC, 2011; 3, FELASA Guidelines and Recommendations; 4, Directive 2010/63/EU; and 5, National Centre for the replacement, refinement and reduction of animals in research recommendations (NC3Rs).

In the case where only plasma and no tissue samples were collected, the animals were killed humanely by using $CO_2$, followed by cervical dislocation as described in the animal ethics approval held by Admescope. When both plasma and tissue samples were collected, the animals were killed under isoflurane anaesthesia via cervical dislocation.

We used male BALB/c mice because this inbred strain of mice as it is commonly used for preclinical pharmacokinetic studies [23]. Mice are commonly used because of their short lifespan, allowing for the growth of a large number of animals in a short period of time and, consequently, the feasibility of many studies to predict pharmacokinetic profiles in humans [24]. We used male mice as this is the sex that is most commonly used; however, we note that sex differences have been reported [25] and this could be investigated in the future for N-acetyl-leucine. Pharmacokinetic data obtained from mice can be used to extrapolate to humans through allometric scaling, but differences between specific drugs require empirical studies in humans [26].

The animals were weighed on the day prior to dosing. The compound was administered to male BALB/c mice (n = 3 per time point) p.o. (100 mg/kg; 10 mL/kg) by oral gavage. Blood samples were collected into potassium EDTA tubes by venepuncture from the saphenous vein. Within 30 min following the sampling, blood was centrifuged for plasma separation (room temperature; 10 min; 2700 x*g*). The plasma samples were transferred into plastic tubes, frozen and stored at –20˚C.

## Chemicals and suppliers

HPLC grade methanol and acetonitrile were from Merck (Darmstadt, Germany). HPLC grade formic acid, acetic acid and ammonium formate were from BDH Laboratory Supplies (Poole, UK). Other chemicals were from Sigma Aldrich (Helsinki, Finland), and of the highest purity available. Water was from a Direct-Q3 (Millipore Oy, Espoo, Finland) purification system and UP grade (ultrapure, 18.2 MW). N-acetyl-DL-leucine was obtained from Molekula (#73891210) and N-acetyl-L-leucine was obtained from Sigma Aldrich (#441511).

## Sample preparation

The plasma samples were prepared for analysis by mixing 50 μL of plasma with 100 μL of acetonitrile and mixed. In addition to plasma, skeletal muscle and brain tissue was also taken at less frequent time points (0.5, 2, 6, 24 and 48 h) and prepared for analysis by homogenization of 50 mg of tissue with 100 μL of acetonitrile. The samples were transferred to Waters 96-well plate and the sample was evaporated under nitrogen gas flow. The sample was reconstituted into 150 μl of 50% methanol:water and analysed by LC/MS. Standard plasma samples were prepared by spiking the injection solution with concentrations from 1 to 10 000 ng/mL by using one volume of spiking solution and nine volumes of injection solution. These samples were then prepared for analysis in the same way as the samples. Quality control (QC) samples were prepared both from racemic-N-Acetyl-Leucine and from N-Acetyl-L-Leucine in two different concentrations. QC samples from racemic-N-Acetyl-Leucine were prepared into concentrations of 40, 400 and 4000 ng/mL, corresponding to 20, 200 and 200 ng/mL concentration of both D- and L-enantiomers, respectively. QC samples of N-Acetyl-L-Leucine was then prepared into concentrations of 20, 200 and 2000 ng/mL. QC samples were then prepared for analysis in the same way as the samples.

## Quantification with liquid chromatography-mass spectrometry and chiral-HPLC

Quantification by HPLC was performed using a Supelco Astec CHIROBIOTIC T chiral HPLC column (2.1 x 150 mm, 5 μm particle size) with a Waters Acquity UPLC + Thermo Q-Exactive hybrid Orbitrap MS, using ESI negative polarity, nitrogen auxiliary gas (450˚C), capillary voltage was 2000 and 350˚C and controlled with the software Xcalibur 4.1. Samples were injected as a 4-μL volume and eluted with a gradient of buffer A (20 mM ammonium acetate) and buffer B (methanol) with a flow rate of 0.3 mL/min and column oven temperature of 30˚C. The gradient was 80% A at 0 min; 20% A at 3.5 min and 80% A at 4.5 min Parallel Reaction Monitoring (PRM) and Full-MS-dd-MS2 were measured at the same time. In PRM, quadrupole was used as a mass filter and depending whether deuterated or non-deuterated N-Acetyl-L-Leucine was detected, either m/z 172 or 176 only got through. Ions with aforementioned m/z was then collided and leucine fragment (m/z 130 or 134) was used in quantitation. In full-MS-dd-MS2 mode, every ion with intensity over a certain intensity was collided and fragments analyzed.

## Metabolite identification with reverse phase ultrahigh performance liquid chromatography

Metabolite identification was performed using a Waters Acquity UPLC + Thermo Q-Exactive hybrid Orbitrap MS and a Waters Acquity HSS T3 column (50 x 2.1 mm, 1.8 μm particle size). MS was as described above over the mass range of 70–1000 using an acquisition time of 7 Hz for full scan, IT 100 ms for DDI MS/MS, an AGC Target of 1E6, maximum IT of 100 ms and 35 000 (FWHM @ m/z 200) for full scan, 17 500 for MS/MS in DDI mode off for full scan; 20 +40+60 for DDI MS/MS inclusion list for expected metabolites ON; also other unexpected most abundant metabolites chosen for MS/MS. Samples were injected as a 4-μL volume and eluted at a flow rate of 0.5 mL/min and a column oven temperature of 35˚C with a gradient consisting of Buffer A 0.1% formic acid and Buffer B acetonitrile. The gradient was (min, %A): 0, 98; 0.5, 98; 2, 50; 3, 5 and 3.5, 5. Ion chromatograms were extracted from the total ion chromatograms using calculated monoisotopic accurate masses with 10 mDa window. The metabolites were mined from the data using software-aided data processing (Thermo Compound Discoverer 2.0 including structure-intelligent dealkylation tool & mass defect filter) with manual confirmation.

## Pharmacokinetic calculations

Plasma pharmacokinetic parameters of the N-acetyl-leucine enantiomers were calculated using Phoenix 64 (Build 6.4.0.768) WinNonlin (version 6.4) software, using non-compartmental method with sparse sampling. Nominal doses were used for all animals. The terminal phase half-life ($T_{1/2}$), the time for 50% of the plasma concentration to decrease after some point of elimination, was calculated by least-squares regression analysis of the terminal linear part of the log concentration–time curve using the relationship $0.693/k_e$. The area under the plasma concentration–time curve (AUC), an estimation of plasma drug exposure over time, was determined with the linear trapezoidal rule for increasing values and log trapezoidal rule for decreasing values up to the last measurable concentration ($AUC_{0-last}$). The first order elimination rate constant $k_e$ was calculated as the slope (minimum 3 points) from the terminal log plasma concentration time curve. The maximum concentration ($C_{max}$) and the time taken to achieve the peak concentration ($T_{max}$) after oral dose were obtained directly from the plasma concentration data without interpolation. The theoretical background and interpretation of the pharmacokinetic data was based on [23]. Where appropriate, data are expressed as the mean ± standard error of the mean. Means were statistically analysed by either pre-planned t tests or a one-sample t test comparing the measured value with the expected value. Graphs were plotted using Prism 7 (GraphPad Software Inc) and organized and formatted in Illustrator (Adobe Inc).

# Results

## N-acetyl-D-leucine exhibits larger $C_{max}$ and AUC following racemate administration

To determine whether the enantiomers of N-acetyl-leucine have different pharmacokinetics, we orally dosed mice with either a racemate or the L-enantiomer (Fig 2 and S1 Fig). Following an oral dose of N-acetyl-DL-leucine (100 mg/kg and 10 mg/mL), in the plasma, the concentration of the D-enantiomer was greater than the L-enantiomer at all time points (Fig 3A). Note that direct comparison of the enantiomers is shown in replots of these data (Fig 4) and will be discussed below. This asymmetry in the plasma concentrations of the D- and L-enantiomers can be quantitated by comparing, respectively, $C_{max}$ of 86100 ng/mL verses 341 ng/mL (Fig 5A

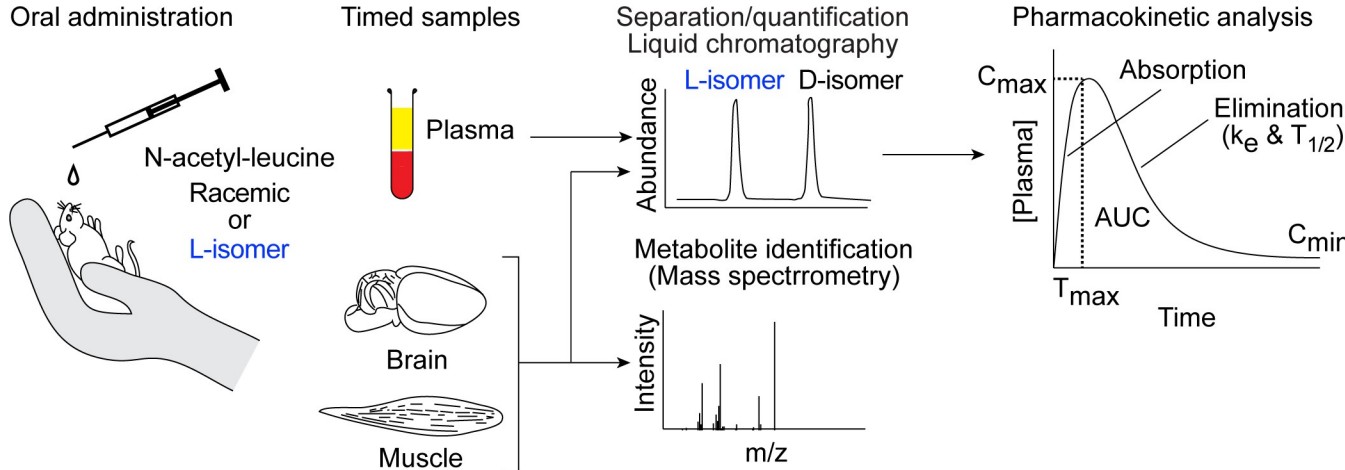

**Fig 2. Schematic outlining the experimental procedure.** Male mice were orally administered N-acetyl-leucine as either the racemate (50% each enantiomer) or purified L-enantiomer (2.6% D-enantiomer and 97.4% L-enantiomer). At specific times (0.25 to 8 h) after administration, blood was taken, plasma was separated and quantified by chiral liquid chromatography/mass spectrometry. Plots of the plasma concentration of each enantiomer over time were used to visualize pharmacokinetics and a noncompartmental model was used to calculate the pharmacokinetic parameters $C_{max}$ (maximum peak concentration), $T_{max}$ (time to reach $C_{max}$), $k_e$ (first order elimination rate constant), $T_{1/2}$ (half-life) and AUC (area under the curve). Samples of brain and skeletal muscle were also taken at specific times and used to determine compound distribution and to search for metabolites with high-resolution mass spectrometry.

and Table 1) and AUC of 75800 h*ng/mL versus 2560 h*ng/mL (Fig 5D and Table 1). The elimination rate was similar for both enantiomers, indicated by the linear and parallel curves on a semilog graph (Fig 3B) using a noncompartmental model giving a $k_e$ of 2.2 h$^{-1}$ for the D-enantiomer and 2.8$^{-1}$ h for L-enantiomer (Fig 5C and Table 1), with corresponding $T_{1/2}$ values of 0.31 h and 0.40 h (Fig 5E and Table 1). The D-enantiomer remained detectable until 8 h

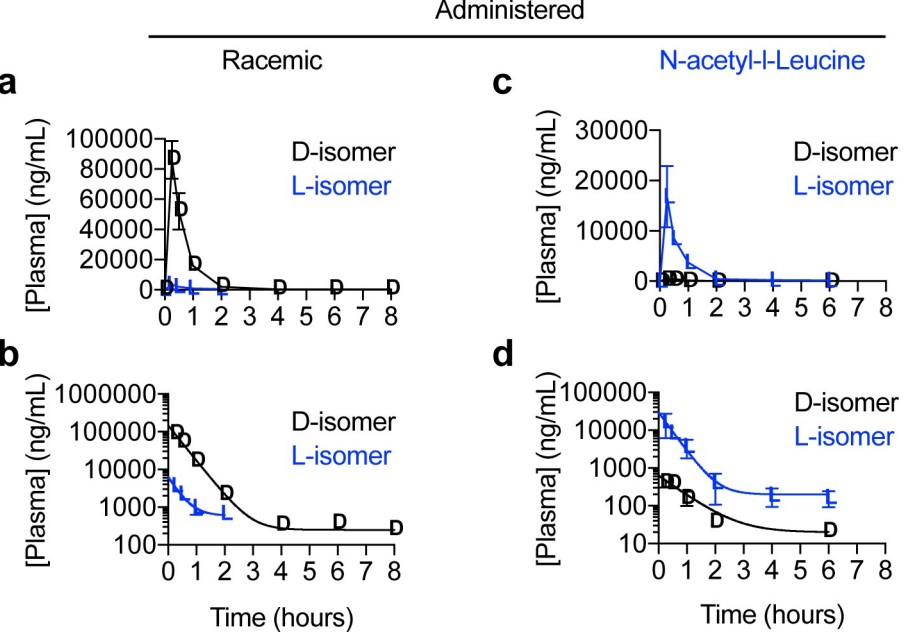

**Fig 3. Graphs of plasma concentration of enantiomers versus time after administration of racemic N-acetyl-DL-leucine or purified N-acetyl-L-leucine.** Data are presented as (**a,c**) linear-linear plots or (**b,d**) semilog plots. Values are the mean ± standard error of the mean with n = 3 (mice).

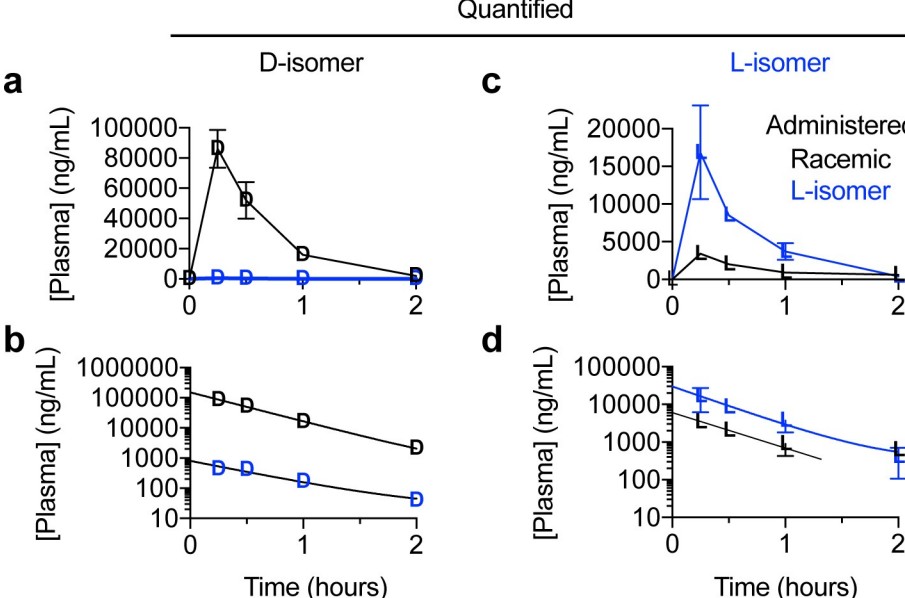

**Fig 4. Replots of the data to facilitate direct comparison of the plasma concentration of N-acetyl-leucine enantiomers after oral administration of racemic N-acetyl-DL-leucine or purified N-acetyl-L-leucine.** Data are presented as (**a,c**) linear-linear plots or (**b,d**) semilog plots. Values are the mean ± standard error of the mean with n = 3 (mice).

(Fig 5G) with the last concentration of 247 ng/mL (Fig 5F). In contrast, the L-enantiomer remained detectable until 2 h (Fig 5G) with the last concentration of 623 ng/mL (Fig 5F).

## Pharmacokinetics of the enantiomers following N-acetyl-L-leucine administration

For oral dosing with purified N-acetyl-L-leucine, the commercial source of this was found to contain 97.4% L-enantiomer and 2.6% of the D-enantiomer (S1 Fig). This trace contamination enabled us to evaluate the pharmacokinetics of the D-enantiomer at a much lower dose, and allowed for an internal control and comparator. Following an oral dose of the purified L-enantiomer at 100 mg/kg and 10 mg/mL, the concentration of the L-enantiomer was greater at all time points (Fig 3C). Quantitatively, for the D- and L-enantiomers, respectively, had a $C_{max}$ of 436 ng/mL versus 16900 ng/mL (Fig 5A and Table 1) and an AUC of 573 h*ng/mL and 11400 h*ng/mL (Fig 5D and Table 1). As with administration of the racemate (Fig 3A and 3B), after dosing with purified L-enantiomer, the elimination rate was similar for both enantiomers, indicated by the linear and parallel curves on a semilog graph (Fig 3D) and was well-fit with a non-compartmental model giving a $k_e$ of 1.7 h$^{-1}$ for the D-enantiomer and 2.4$^{-1}$ for L-enantiomer (Fig 5C and Table 1), with corresponding $T_{1/2}$ values of 0.25 h and 0.29 h (Fig 5E and Table 1). Both enantiomers remained detectable in the plasma until 8 h and 6 h (Fig 5G) with a last concentration of 16 ng/mL and 168 ng/mL (Fig 5F). However, the $C_{last}$ and $T_{last}$ are somewhat misleading for all measurements, as in looking at the profiles, the main elimination was over for all enantiomers at around 4 h when these terminal concentrations were reached (Fig 3B and 3D).

## Dose proportionality is greater than unity

Dose proportionality refers to the effect of an increase in dose on $C_{max}$ and AUC [23]. We can assess dose proportionality with our data by using the amount of each enantiomer present in

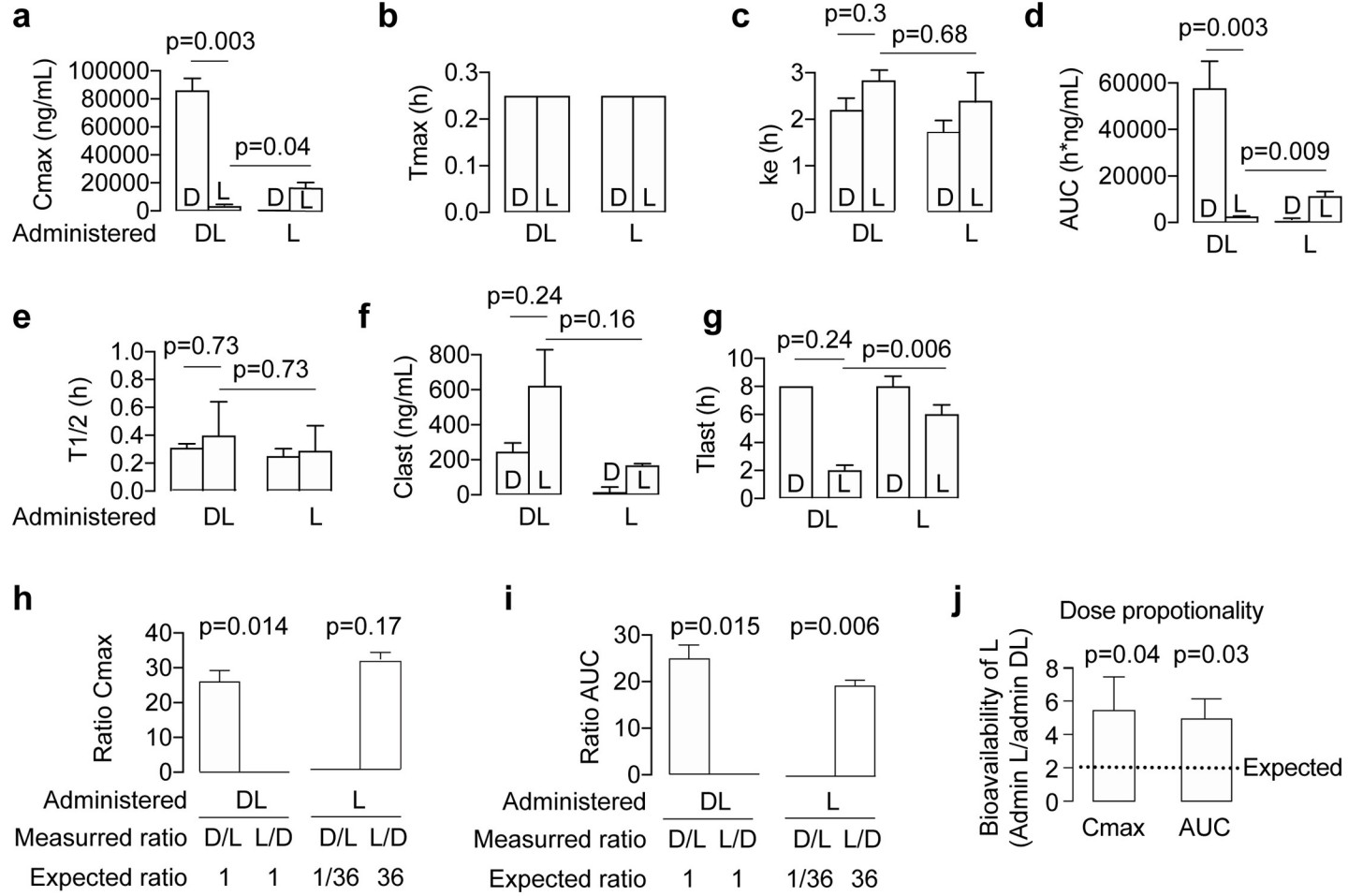

**Fig 5. Bar charts showing the pharmacokinetic parameters for the enantiomers of N-acetyl-L-leucine after administration of racemic N-acetyl-DL-leucine (denoted as DL) or N-acetyl-L-leucine (denoted as L).** (**a-g**) Conventional pharmacokinetic parameters calculated from the plasma concentration of drug. (**h-j**) Parameters derived from the conventional pharmacokinetic parameters to detect and highlight the effects of pharmacokinetic differences between the enantiomers. Values are the mean ± standard error of the mean with n = 3 (mice). Means were statistically analysed by either (**a-g**) pre-planned t tests; (**h and i**) a one-sample t test comparing the measured value with the expected value: 1 when administered as DL and 36 when administered as purified L; and (**j**) a one-sample t test comparing the measured value with the expected value of 2. The means compared are indicated by the horizontal lines on the charts, and exact p values are provided for the comparisons.

the composition administered. The D-enantiomer was dosed as 50% of the administered racemate and as 2.6% of the administered purified L-enantiomer, for a difference in dose proportionality of 19-fold. The actual dose proportionality was 197-fold for $C_{max}$ (86100/436; Table 1) and 101 fold for AUC (57800/573; Table 1). The L-enantiomer was dosed as 50% of the administered racemate and 97.4% of the administered purified L-enantiomer, for a difference in dose proportionality of 1.9-fold. The actual dose proportionality was 4.9-fold for $C_{max}$ (16800/3410; Table 1 and Fig 5J) and 4.6 fold for AUC (11400/2560; Table 1 and Fig 5J).

## Direct comparison of enantiomers highlights pharmacokinetic differences

To facilitate comparison of the racemate with the purified L-enantiomer, we re-plotted the plasma concentration versus time profiles of the two enantiomers on the same graph over the first two hours (Fig 4). The amount of D-enantiomer in the plasma is significantly higher when dosed with the racemate compared to the much lower amount present when dosed with

**Table 1. The calculated pharmacokinetic parameters for N-Acetyl-D-Leucine and N-Acetyl-L-Leucine plasma after oral administration of N-Acetyl-DL-Leucine or N-Acetyl-L-Leucine at a nominal dose of 100 mg/kg.**

| Compound administered | | N-Acetyl-DL-Leucine | | N-Acetyl-L-Leucine | |
| --- | --- | --- | --- | --- | --- |
| Compound quantified | | N-Acetyl-D-Leucine | N-Acetyl-L-Leucine | N-Acetyl-D-Leucine | N-Acetyl-L-Leucine |
| Parameter | Unit | Value | Value | Value | Value |
| $r^2$ | - | 0.91 | 0.93 | 0.85 | 0.72 |
| $k_e$ | - | 2.2 | 2.8 | 1.7 | 2.4 |
| $T_{max}$ | h | <0.25 | <0.25 | <0.25 | <0.25 |
| $C_{max}$ | ng/mL | 86 100 | 3410 | 436 | 16 800 |
| $T_{last}$ | h | 8.00 | 2.00 | 8.00 | 6.00 |
| $C_{last}$ | ng/mL | 247 | 623 | 16.2 | 168 |
| $T_{1/2}$ | h | 0.31 | 0.4 | 0.25 | 0.29 |
| $AUC_{0\text{-}last}$ | h x ng/mL | 57 800 | 2 560 | 573 | 11 400 |
| Ratio $C_{max}$ L/D [#] | - | 0.04 | | 38.5 | |
| Ratio AUC L/D [#] | - | 0.04 | | 19.8 | |

[#]The ratio of corresponding value between L and D enantiomers

purified L-enantiomer, and is consistent with the measured 2.6% D contamination in the purified L-enantiomer (S1 Fig). The semilog plot nicely shows the equal rates of elimination at all concentrations and times, demonstrating that the D-enantiomer is not affected by dosing with either the DL or L form. As would be expected with administering a 97.4% to 2.6% mixture of N-acetyl-L-leucine to N-acetyl-D-leucine, the L-enantiomer dominated in the plasma (Fig 4A and 4B). Administration of DL or L alone only affected $C_{max}$ and AUC, but did not affect elimination ($k_e$ o $T_{1/2}$). Plotting the L-enantiomer in the plasma on the same graph to compare dosing with DL with L alone (Fig 4C), graphically shows the dramatic differences in $C_{max}$ and AUC, but show the same rate of elimination (parallel curves when fit to a noncompartmental model).

Another way to compare administration of the racemate to the purified L-enantiomer on the pharmacokinetics of the enantiomers was to calculate the ratio of enantiomers in regard to $C_{max}$ and AUC. As we verified the administered compound to be a true racemate (50% each enantiomer; S1 Fig), deviations from a ratio of 1 reveal significantly different pharmacokinetics between the D- and L-enantiomers. When administered as the racemate, the ratio of D/L enantiomer was about 25 for both $C_{max}$ (Fig 5H; 26 versus 1, p = 0.014) and AUC (Fig 5I, 25 vs 1, p = 0.015). As the purified L-enantiomer administered contained 97.4% L-enantiomer and 2.6% D-enantiomer (S1 Fig), if the enantiomers had identical pharmacokinetics, the ratio of L/D would be predicted to be 36 (that is, 97.4/2.6). When administered as the purified L-enantiomer, the ratio of L/D was 32 for $C_{max}$ (Fig 5H; 31.7 versus 36, *p* = 0.17) and 20 for AUC (Fig 5I; 19.8 versus 36, *p* = 0.006).

## Enantiomers show differences in distribution and metabolism

To investigate the effect of administering either the racemate or purified L-enantiomer of N-acetyl-leucine on the distribution of the enantiomers, muscle and brain were analysed. At specific times after oral dosing, the mice were euthanized and the amount of D- and L-enantiomer present in the tissues was determined. Following oral dosing with the racemate, muscle contained much more D-enantiomer than L-enantiomer (Fig 6A). In muscle, the D-enantiomer was only detectable at 30 min and 2 h (Fig 6A). In contrast, following oral dosing of the L-enantiomer alone, in muscle, the L-enantiomer was not detected at any time point and the D-

enantiomer was detectable but at a much lower concentration (Fig 6C) than after administration of the racemate (Fig 6A). Neither the D- nor L-enantiomer was detected in muscle after 2 hours from the time or dosing (Fig 6A and 6C). Following oral dosing with the racemate, the brain contained detectable D-enantiomer at only the 30 min time point and L-enantiomer was not detectable at any time point (Fig 6B). Following oral dosing with purified L-enantiomer, neither of the enantiomers were detected at any time point (Fig 6D).

We investigated the identity of the metabolites of both enantiomers in muscle, but no metabolites of either N-acetyl-D-leucine or N-acetyl-L-leucine were detected (data not shown).

## Discussion

We investigated the pharmacokinetics of the enantiomers of N-acetyl-leucine after oral administration of the racemate, which has been marketed under the name Tanganil® for the treatment of vertigo in France since 1957 [1], and the purified L-enantiomer, which is the pharmacologically active enantiomer in models of acute vertigo [4,19]. We report significant and unexpected differences in the pharmacokinetics of the enantiomers. The major findings of this study are as follows: First, when administered as the racemate (N-acetyl-DL-leucine), the D-enantiomer was present at much higher plasma maximal concentration ($C_{max}$) and (area under the curve; AUC) relative to the L-enantiomer, resulting in greater total exposure. Second, when administered as purified N-acetyl-L-leucine, both the $C_{max}$ and the AUC for N-acetyl-L-leucine were higher compared to administration as the racemate, even when scaled for the relative dose. Third, both enantiomers distributed to the tissues monitored, muscle and brain, but the D-enantiomer was found at much higher concentrations relative to the L-enantiomer in both tissues.

### Origin of the differences in $C_{max}$ and AUC

The larger AUC for N-acetyl-L-leucine when administered as the purified enantiomer compared to when administered as the racemate, and factoring in the actual amounts of L-enantiomer present in each (that is, 97.6% and 50%, respectively), is fully accounted for the by increase in $C_{max}$ because after $C_{max}$ and $T_{max}$, the clearance ($k_e$ and $T_{1/2}$) is the same for both enantiomers. In other words, after 15 min, the pharmacokinetic parameters are the same for both enantiomers. Therefore, the large differences in $C_{max}$ have to be due to processes occurring in the first 15 min and before the L-enantiomer enters the plasma. Consequently, we can deduce that the D-enantiomer is interfering with the bioavailability (the amount of drug orally administered that is systemically available) of the L-enantiomer during the first 15 min following oral administration. Differences between enantiomers indicate interaction with protein targets; therefore, two possible explanations that are not mutually exclusive exist: competition at a carrier on cells in the intestine and/or differences in first-pass metabolism.

### Stereoisomer-mediated pharmacokinetics arising from uptake

The bioavailability of a drug is determined by its ability to penetrate and cross the gastrointestinal epithelial cell membrane, either by passive diffusion or via a carrier. Uptake by passive diffusion is determined by physicochemical properties, primarily hydrophobicity, which allows penetration of the membrane's core [27,28]. The N-acetylation of leucine would be predicted to greatly increases passive membrane transport, as it eliminates one ($NH_3^+$) of the two ($NH_3^+$ and $COO^-$) charges present on all amino acids at physiological pH, which can increase transport rates up to $10^{10}$-fold [29,30]. However, as this physicochemical effect (loss of charge and increase in hydrophobicity) is identical for the enantiomers, it cannot underlie the

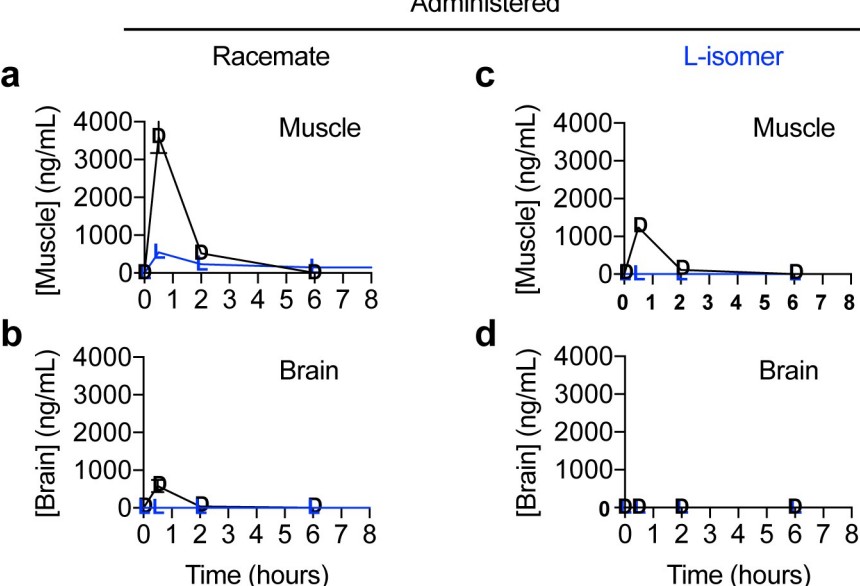

**Fig 6. Graphs of the concentration of enantiomers in tissue versus time after administration of racemic N-acetyl-DL-leucine or purified N-acetyl-L-leucine.** Data are for (**a,b**) muscle and (**c,d**) brain and presented as linear-linear plots. Values are the mean ± standard error of the mean with n = 3 (mice).

differences observed in the pharmacokinetics of the N-acetyl-leucine enantiomers. In contrast, uptake by carriers requires molecular recognition at saturable binding sites and would give rise to interference between the enantiomers. The identity of the carrier for N-acetyl-leucine on the intestinal brush-border membrane is unknown; however, given that N-acetyl-leucine is a modified amino acid, the most likely candidates are amino acid transporters, as 52 families exist that show distinct substrate selectivity[31–33]. These possibilities can be narrowed down further based on the effect of N-acetylation, which forms an amide bond (Fig 1). An amide bond would both make N-acetyl-leucine appear more like a dipeptide and, through resonance, given the C-N bond partial double bond character with a bond order 1.5 [34], making it an analogue of an imine (Fig 1B). These predict that N-acetyl-leucine would be a substrate for the low affinity/high capacity a $H^+$-coupled di/tripeptide transporter termed PepT1, which is highly expressed and responsible for 80% of all amino acids are taken up from the small intestine lumen, or an imino acid transporter which has 100-fold greater affinity for N-modified amino acids and shows only 2-fold stereoselectivity [35].

## Stereoisomer-mediated pharmacokinetics arising from first-pass metabolism

Another likely contributing process accounting for the differences between enantiomers in $C_{max}$ and AUC is first-pass metabolism [21]. As first-pass metabolism is an enzymatic process, it exhibits molecular recognition at saturable binding sites and would also give rise to interference between the enantiomers. Such stereoselective first-pass effects are known to alter oral drug bioavailability of the enantiomers of propranolol and verapamil [22,36]. Indeed, the 2-3-fold stereoisomer effect we detected for N-acetyl-leucine is similar to the 2–3 fold greater oral bioavailability of (–)-verapamil compared to (+)-verapamil caused by first-pass metabolism [21]. Most often first-pass metabolism is mediated by cytochrome P-450 oxidation in the stomach, intestine or liver [21]; however, N-acetyl-L-leucine is more likely handled like a

nutrient than a xenobiotic, as it is a naturally occurring metabolite of L-leucine and a transacetylase has been reported that interconvert N-acetyl-L-leucine and L-leucine, using other L-amino acids as the substrate or product [37,38]. Therefore, a likely enzyme for first-pass metabolism of N-acetyl-L-leucine would be the acylase reported in intestinal strips that was able to remove the acetyl group from most amino acids [39], and showed 40,000-fold selectivity for L-amino acids over D-amino acids [37–40].

## Stereoisomer effects manifested by tissue uptake and metabolism

In regard to the presence of the enantiomers in muscle and brain, the amounts were much lower than in the plasma (10-fold to undetectable), and the D-enantiomer was present at a much higher concentration than the L-enantiomer. In general, our results showing that N-acetyl-leucine is blood-brain barrier permeable are consistent with studies in monkeys in which radioactive racemic N-acetyl-leucine was administered intravenously and radioactivity was subsequently detected in the brains [41]. However, the $^{14}$C label was in the alpha carbon of leucine and autoradiography was used for quantification, so there is no ability to determine whether the radioactivity was due to N-acetyl-DL-leucine itself or a metabolite [41]. Therefore, the data with radioactivity is ambiguous in terms of both the effect of stereoisomerism and whether N-acetylation promotes uptake and whether it is rapidly metabolized to L-leucine.

In contrast to the situation with uptake from the gut to the plasma in which the D-enantiomer was reducing uptake, in muscle and brain, the presence of the D-enantiomer was associated with increased presence of N-acetyl-L-leucine. Uptake from the plasma into cells and tissues, as described for the intestinal cells above, occurs through both passive diffusion and carriers. The explanation of competitive inhibition for a common carrier used for the asymmetry in uptake between the enantiomers into the plasma of competition cannot explain this observation. Indeed, such an effect would result in less of the L-enantiomer, not more, when N-acetyl-D-leucine was also present. A more likely explanation is competitive inhibition of the enantiomers at an enzyme that metabolizes N-acetyl-L-leucine. A likely explanation is that the D-enantiomer is inhibiting the deacetylation of N-acetyl-L-leucine. It is also important to note that the amount of N-acetyl-L-leucine in tissues is a steady state measure of the compound, and relates not to lack of uptake but rather rapid utilization. By comparison, the D-enantiomer was present in higher amounts, consistent with it being metabolically inert based on feeding N-acetyl-D-leucine to rats, where it was excreted in the urine unchanged [38]. The simplest explanation is that the N-acetyl-L-leucine is rapidly converted to L-leucine and utilized in metabolism. Rapid utilization and metabolism of L-leucine is consistent with the results of a study using stable isotope-labelled leucine itself upon oral administration [42]. Moreover, our inability to detect metabolites is consistent with the disappearance of N-acetyl-L-leucine though metabolism to L-leucine, which would be undetectable on the background of endogenous L-leucine. Slowing the conversion of N-acetyl-L-leucine to L-leucine, and subsequently its regulatory effect on muscle protein synthesis and oxidative metabolism [43,44], and possibly impact on its efficacy as a drug. Taken together, these data showing low amounts of N-acetyl-leucine in the brain and muscle suggest that the mechanism of action of N-acetyl-L-leucine requires metabolism.

## Clinical implications of stereoselective pharmacokinetics

The different pharmacokinetics of the enantiomers would conceivably result in disproportionate total exposure (increase in the AUC) to the D-enantiomer when the racemate is dosed, as the L-enantiomer would be eliminated much faster. Importantly, chronic treatment with multiple doses over time would cause accumulation in the body of the D-enantiomer of N-acetyl-

leucine. Historically, it was presumed that the 'inactive' enantiomer was harmless [16], a notion disabused by the thalidomide tragedy [15]. Although the N-acetyl-D-leucine is not reported to be toxic, concerns about the toxicity of D-amino acids in general have been raised as the reason for the original evolutionary selection and biological presence of D-amino acid oxidase [45,46]. Evidence that the D-leucine is having a biological effect comes from a report in which low amounts (about 1/10th of endogenous L-form) of D-leucine suppressed endogenous levels of L-leucine by almost half [47].

## Conclusions

In conclusion, firstly, the L-enantiomer–which is the pharmacologically active form in models of acute vertigo–has different pharmacokinetics when administered with the D-enantiomer as the racemate (N-acetyl-DL-leucine) compared to administration as the purified L-enantiomer. Secondly, we found evidence for an accumulation of the D-enantiomer, which would be exacerbated by chronic dosing of the racemate, with unknown and possibly unwanted deleterious effects on cell function. Thirdly, the results of this study, taken together with the regulatory guidelines of the FDA [17] and the EMA [18], strongly supports the research and development of isolated N-acetyl-L-leucine.

## Supporting information

**S1 Fig. Chiral high performance liquid chromatography/mass spectrometry analysis showing separation and quantification of the compounds used in these studies.** (**a**) Spectrum of racemate. (**b**) Spectrum of purified N-acetyl-L-leucine. Note that the peak areas are not directly comparable with concentration due to differences in the extent of ionization of the compound in the mass spectrometers ionization chamber due to relative differences in aqueous and organic solvent concentrations in the mobile phase at those time points due to a gradient elution. Therefore, quantification was based on a standard curve specific to each enantiomer. The result is that the racemate contained 50.2% N-acetyl-D-leucine and 49.8% N-acetyl-L-leucine. The purified N-acetyl-L-leucine contained 2.6% N-acetyl-D-leucine and 97.4% N-acetyl-L-leucine. The limit of detection and the limit of quantification was, respectively, 10 ng/mL and 25 ng/mL for N-acetyl-D-leucine, and 25 ng/mL and 50 ng/mL for N-acetyl-L-leucine. (TIF)

**S1 Table. Measured concentrations of N-Acetyl-L-Leucine and N-Acetyl-D-Leucine in mouse plasma and tissues after p.o administration N-Acetyl-DL-Leucine at 100 mg/kg.** (DOCX)

**S2 Table. Measured concentrations of N-Acetyl-L-Leucine and N-Acetyl-D-Leucine in mouse plasma and tissues after p.o administration N-Acetyl-L-Leucine at 100 mg/kg.** (DOCX)

## Author Contributions

**Conceptualization:** Grant C. Churchill, Antony Galione, Frances M. Platt.

**Data curation:** Grant C. Churchill.

**Formal analysis:** Grant C. Churchill.

**Funding acquisition:** Grant C. Churchill, Antony Galione, Frances M. Platt.

**Methodology:** Grant C. Churchill, Antony Galione, Frances M. Platt.

**Visualization:** Grant C. Churchill.

**Writing – original draft:** Grant C. Churchill.

**Writing – review & editing:** Michael Strupp, Antony Galione, Frances M. Platt.

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
