## [Decision Letter · Decision Letter 0]

27 Sep 2019

PONE-D-19-24169

Unexpected differences in the pharmacokinetics of N-acetyl DL-leucine enantiomers after oral dosing and their clinical relevance

PLOS ONE

Dear Dr Churchill,

Thank you for submitting your manuscript to PLOS ONE. After careful consideration, we feel that it has merit but does not fully meet PLOS ONE’s publication criteria as it currently stands. Therefore, we invite you to submit a revised version of the manuscript that addresses the points raised during the review process.

We would appreciate receiving your revised manuscript by Nov 11 2019 11:59PM. To enhance the reproducibility of your results, we recommend that if applicable you deposit your laboratory protocols in protocols.io, where a protocol can be assigned its own identifier (DOI) such that it can be cited independently in the future. For instructions see: http://journals.plos.org/plosone/s/submission-guidelines#loc-laboratory-protocols

We look forward to receiving your revised manuscript.

Kind regards,

Nicolás Pérez-Fernández

Academic Editor

PLOS ONE

Journal Requirements:

2. At this time, we request that you please report additional details in your Methods section regarding animal care, as per our editorial guidelines. In the cases where mice were euthanized using CO2 asphyxiation (no samples collected), please state whether a secondary method of euthanasia was used to ensure the humane sacrifice of the mice.

"I have read the journal's policy and the authors of this manuscript have the following competing interests: MS is Joint Chief Editor of the Journal of Neurology, Editor in Chief of Frontiers of Neuro-otology and Section Editor of F1000. He has received speaker’s honoraria from Abbott, Actelion, Auris Medical, Biogen, Eisai, Grünenthal, GSK, Henning Pharma, Interacoustics, Merck, MSD, Otometrics, Pierre-Fabre, TEVA, UCB. He is a shareholder of IntraBio. He acts as a consultant for Abbott, Actelion, AurisMedical, Heel, IntraBio and Sensorion. GCC, AG and FP are cofounders, shareholders and consultants to IntraBio. FP is a consultant to Actelion. IntraBio Ltd is the applicant for patents WO2018229738 (Treatment For Migraine), WO2017182802 (Acetyl-Leucine Or A Pharmaceutically Acceptable Salt Thereof For Improved Mobility And Cognitive Function), WO2019078915 and WO2018029658 (Therapeutic Agents For Neurodegenerative Diseases), WO2018029657 (Pharmaceutical Compositions And Uses Directed To Lysosomal Storage Disorders), and WO2019079536 (Therapeutic Agents For Improved Mobility And Cognitive Function And For Treating Neurodegenerative Diseases And Lysosomal Storage Disorders)."

Reviewers' comments:

Reviewer's Responses to Questions

**Comments to the Author**

1. Is the manuscript technically sound, and do the data support the conclusions?

Reviewer #1: Yes

Reviewer #2: Yes

2. Has the statistical analysis been performed appropriately and rigorously? 

Reviewer #1: Yes

Reviewer #2: Yes

3. Have the authors made all data underlying the findings in their manuscript fully available?

Reviewer #1: No

Reviewer #2: Yes

4. Is the manuscript presented in an intelligible fashion and written in standard English?

Reviewer #1: Yes

Reviewer #2: Yes

5. Review Comments to the Author

Reviewer #1: This work study a relevant problem related to the pharmacokinetics of N-acetyl DL-leucine enantiomers that has been used for vertigo treatment as an over the counter drug, essentially in France. The results suggest that during chronic administration of the racemate, the D enantiomer accumulate inducing negative effects. When N-acetyl L-leucine is administered the dose proportionality suggests saturable processes affecting uptake and metabolism. These results are explained by inhibition of uptake at an intestinal carrier of the L-enantiomer by the D-enantiomer, and by metabolism of the L-, but not D-enantiomer, likely by deacetylation. The enantiomers of N-acetyl-leucine exhibit large, unexpected differences in pharmacokinetics supporting the use of N-acetyl-L-leucine instead of the racemate or N-acetyl-D-leucine.

Introduction provides scarce information regarding the use and actions of N-acetyl DL-leucine, some considerations about its potential mechanism of action for the vertigo treatment will be welcome by readers.

Diverse corrections particularly reorganization of the manuscript and figures is needed.

Punutal observations

Rows 66-67. This statement is true only for vertebrates, the insects have essentially D amino-acids so please in the sake of knowledge introduce this clarification

Rows 140-141. Please be more specific. In which sense do recording of clinical signs or behaviors of animals minimize suffering?

Row 153 In which case the tissue was not collected from animals. Please clarify this point.

Row 159. Sample size is minimal acceptable thus statistical potency of the results low.

Since repeated measurements along time were performed and only 3 data for each point a further analysis about statistical significance should be included, also authors should consider use of other tests such as repeated measurements ANOVA and post hoc tests.

More details about experimental design are needed since authors did not mention any positive or negative controls, also information about why they decided those timings for the samplings of the drug and also information about who they handled the tissues of the animal is needed.

Authors refer to plasma samples but not tissue samples. So it is not clear at all who they measured and compared brain and muscle concentrations of the drug. Only in lines 265-269 some mention of this is done.

Figure 4 is referred after figure 5 in the text. Please invert the order.

Line 398, seems something is lacking "accounted for the by increase in Cmax"

Figure 3. The use of letters as symbols in the graphs is confusing and do not contribute to figure compression in fact it is cumbersome and did not allow comparison of the curves (same for figure 4).

figure 5 What is being compared in h, i and j.

Figure 5. What is being compared in h, i and j?

In figure 6 in Y Axes Use mg instead of ng

Use of letters for symbols in the graphs seems not adequate it makes impossible to read value in brain for L and symbols are one in top of the other. Maybe use of bars would be adequate to rearrange this graphs and the whole figure.

Reviewer #2: The manuscript titled “Unexpected differences in the pharmacokinetics of N-acetyl DL-leucine enantiomers after oral dosing and their clinical relevance” by Strupp and colleagues details the pharmacokinetics of the racemate (N-acetyl-DL-leucine) or the nearly purified active L-enantiomer (N-acetyl-L-leucine) in mice. N-acetyl-leucine, as a racemic mixture, has been used as an over the counter anti-vertiginous drug and has also more recently been utilized in cerebellar ataxia, migraine, lysosomal storage disorders, and with age-related neural issues in the elderly. Recent evidence in animal models have indicated that the effectiveness of N-acetyl-DL-leucine is likely attributed to the L-enantiomer, N-acetyl-L-leucine. However, much is still to be worked out regarding the mechanism of action, pharmacodynamics, and particularly pharmacokinetics – all aspects of modern-day drug development that would further facilitate the use of N-acetyl-L-leucine or related analogs for treatment of the aforementioned medical issues, as well as possible improve their overall effectiveness. The data in this manuscript convincingly demonstrate that the pharmacologically active form N-acetyl-L-leucine has fundamentally different pharmacokinetics when given as a 50:50 racemate (N-acetyl-507 DL-leucine) when compared to administration of purified preparation containing mostly L-enantiomer and small amounts of the D-isomer. The pharmacokinetic results are consistent with probable inhibition of intestinal uptake of N-acetyl-L-leucine by N-acetyl-D-leucine in conjunction with selective first-pass metabolism of N-acetyl-L-leucine. Low levels of N-acetyl-L-leucine in the brain and muscle, even in the case of N-acetyl-L-leucine administration alone, suggest that N-acetyl-L-leucine may work as a prodrug that gives rise to the active agent upon metabolism. That metabolite is currently unidentified. The paper is reasonably well written and the data are statistically sound and clearly presented. Some statements need to be rephrased to improve clarity. I have listed only one moderate concern and several minor concerns below:

Moderate concern:

What is the rationale or motivation for using Balb/c mice? In light of recent concerns about sex as a biological factor, why were males mice only included in this study? How applicable are the pharmacokinetic data in mice to that in humans? Consider elaborating some of these details in the material and methods and/or discussion.

Minor issues:

Abstract as it appears in the main body of the manuscript is incomplete in the abstract box on the title page. The first few sentences are missing. Not sure if this is was a post-uploading issue or inadvertent deletion from the authors.

Abstract, pg 2, ln 49: Consider changing “…support the research and development of isolated N-acetyl-L-leucine” to “support the research and development of using only N-acetyl-L-leucine”.

Introduction, pg 3, Ln 55: Consider changing “as it is a” to “as a”.

Introduction, pg 3, Ln 56: Consider changing “cerebella” to “cerebellar”.

Introduction, pg 3, Ln 61: For clarity, consider rewriting “As N-acetyl-leucine is an analogue of the alpha amino acid leucine, it has a stereocentre and thus a pair of enantiomers” to “N-acetyl-leucine, an analogue of the alpha amino acid leucine, has a stereocentre and thus a pair of enantiomers”.

Introduction, pg 4, Ln 80: Consider changing “…from nice to know to effectively need

to know for informed dosing…” to “…from a “nice to know” to effectively “need

to know” basis for informed dosing…”

Introduction, pg 4, Ln 81-82: Consider simplifying “…it was and continues to be marketed as a racemate…” to ““…it was and continues to be used and marketed as a racemate…”

Introduction, pg 4, Ln 88-89: Consider inserting that and at in the following sentence: “each with distinct properties with one (N-acetyl-D-leucine) at best that does not contribute to the therapeutic response, and at worst is potentially responsible for toxicity, as this inactive enantiomer provides ‘therapeutic ballast’

Please elaborate on what you mean with the term “therapeutic ballast”. Is that synonymous with isomeric ballast?

Introduction, pg 4, Ln 96: Should “pharmacokinetic” be “pharmacokinetics”?

MandM, pg 6, ln 141: Should “…distress, clinical signs and general behaviour of the

animals was recorded…” be “distress, clinical signs and general behaviour of the

141 animals were recorded…”

MandM, pg 7, ln 153: “where no (ssue samples…” should be “where no tissue samples".

MandM, pg 11, ln 248: Consider changing “quantitated” to “quantified”.

Results, pg 13, ln 302: Consider rewriting “the main elimination was over for all enantiomers at around 2 h” as the elimination for the D-isomer is closer to four hours.

Results, pg 13, ln 326: “was the plasma is significantly higher” to “in the plasma is significantly higher”.

Results, pg 15, ln 344: delete “compared” from “Another way to compare administration of the racemate compared to the purified enantiomer…” to read “Another way to compare administration of the racemate to the purified enantiomer…”

Consider adding superscript asterisks to table to indicate level of significance on values where statistical tests were performed and are relevant.

6. PLOS authors have the option to publish the peer review history of their article (what does this mean?). If published, this will include your full peer review and any attached files.

Reviewer #1: No

Reviewer #2: No

---

## [Author Response · Author response to Decision Letter 0]

28 Jan 2020

PONE-D-19-24169

Unexpected differences in the pharmacokinetics of N-acetyl DL-leucine enantiomers after oral dosing and their clinical relevance

PLOS ONE

Dear Dr Churchill,

Thank you for submitting your manuscript to PLOS ONE. After careful consideration, we feel that it has merit but does not fully meet PLOS ONE’s publication criteria as it currently stands. Therefore, we invite you to submit a revised version of the manuscript that addresses the points raised during the review process.

We would appreciate receiving your revised manuscript by Nov 11 2019 11:59PM. To enhance the reproducibility of your results, we recommend that if applicable you deposit your laboratory protocols in protocols.io, where a protocol can be assigned its own identifier (DOI) such that it can be cited independently in the future. For instructions see: http://journals.plos.org/plosone/s/submission-guidelines#loc-laboratory-protocols

• A rebuttal letter that responds to each point raised by the academic editor and reviewer(s). This letter should be uploaded as separate file and labeled 'Response to Reviewers'.

• A marked-up copy of your manuscript that highlights changes made to the original version. This file should be uploaded as separate file and labeled 'Revised Manuscript with Track Changes'.

• An unmarked version of your revised paper without tracked changes. This file should be uploaded as separate file and labeled 'Manuscript'.

We look forward to receiving your revised manuscript.

Kind regards,

Nicolás Pérez-Fernández

Academic Editor

PLOS ONE

Journal Requirements:

Response: We confirm that our manuscript meets PLOS ONE’s style requirements.

2. At this time, we request that you please report additional details in your Methods section regarding animal care, as per our editorial guidelines. In the cases where mice were euthanized using CO2 asphyxiation (no samples collected), please state whether a secondary method of euthanasia was used to ensure the humane sacrifice of the mice.

Response: We have reworded the section to clarify our meaning and added that cervical dislocation was also used for these animals. We have added this in lines 190-192 in the revised manuscript.

"I have read the journal's policy and the authors of this manuscript have the following competing interests: MS is Joint Chief Editor of the Journal of Neurology, Editor in Chief of Frontiers of Neuro-otology and Section Editor of F1000. He has received speaker’s honoraria from Abbott, Actelion, Auris Medical, Biogen, Eisai, Grünenthal, GSK, Henning Pharma, Interacoustics, Merck, MSD, Otometrics, Pierre-Fabre, TEVA, UCB. He is a shareholder of IntraBio. He acts as a consultant for Abbott, Actelion, AurisMedical, Heel, IntraBio and Sensorion. GCC, AG and FP are cofounders, shareholders and consultants to IntraBio. FP is a consultant to Actelion. IntraBio Ltd is the applicant for patents WO2018229738 (Treatment For Migraine), WO2017182802 (Acetyl-Leucine Or A Pharmaceutically Acceptable Salt Thereof For Improved Mobility And Cognitive Function), WO2019078915 and WO2018029658 (Therapeutic Agents For Neurodegenerative Diseases), WO2018029657 (Pharmaceutical Compositions And Uses Directed To Lysosomal Storage Disorders), and WO2019079536 (Therapeutic Agents For Improved Mobility And Cognitive Function And For Treating Neurodegenerative Diseases And Lysosomal Storage Disorders)."

Response: We have added the requested statement and included this text in our cover letter.

Response: Our statement complies with PLOS ONE’s policy.

Reviewers' comments:

Reviewer's Responses to Questions

Comments to the Author

1. Is the manuscript technically sound, and do the data support the conclusions?

Reviewer #1: Yes

Reviewer #2: Yes

2. Has the statistical analysis been performed appropriately and rigorously? 

Reviewer #1: Yes

Reviewer #2: Yes

3. Have the authors made all data underlying the findings in their manuscript fully available?

Reviewer #1: No

Reviewer #2: Yes

Response: We have added all the underlying raw data in the form of two tables that are included as Supporting Information (Table S1 and Table S2) in the revised manuscript.

4. Is the manuscript presented in an intelligible fashion and written in standard English?

Reviewer #1: Yes

Reviewer #2: Yes

5. Review Comments to the Author

Reviewer #1: This work study a relevant problem related to the pharmacokinetics of N-acetyl DL-leucine enantiomers that has been used for vertigo treatment as an over the counter drug, essentially in France. The results suggest that during chronic administration of the racemate, the D enantiomer accumulate inducing negative effects. When N-acetyl L-leucine is administered the dose proportionality suggests saturable processes affecting uptake and metabolism. These results are explained by inhibition of uptake at an intestinal carrier of the L-enantiomer by the D-enantiomer, and by metabolism of the L-, but not D-enantiomer, likely by deacetylation. The enantiomers of N-acetyl-leucine exhibit large, unexpected differences in pharmacokinetics supporting the use of N-acetyl-L-leucine instead of the racemate or N-acetyl-D-leucine.

Introduction provides scarce information regarding the use and actions of N-acetyl DL-leucine, some considerations about its potential mechanism of action for the vertigo treatment will be welcome by readers.

Diverse corrections particularly reorganization of the manuscript and figures is needed.

Punutal observations

Response: We have added a section on the potential mechanisms of action of N-acetyl DL-leucine in the treatment of vertigo on lines 54-63 in the revised manuscript.

(Note: We take the word ‘punutal’ to be a typo, but if it has meaning, please clarify.)

Rows 66-67. This statement is true only for vertebrates, the insects have essentially D amino-acids so please in the sake of knowledge introduce this clarification

Response: Although there are free D-amino acids in all organisms that come from diet or metabolism and play roles of toxins or signalling molecules, and certain peptides (defined as chains with < 50 amino acids) such as toxins and hormones contain D-amino acids, we are referring to proteins (defined as chains with > 50 amino acids) synthesized by mRNA. We cannot find any evidence to the contrary that proteins synthesized on ribosomes and encoded by mRNA contain all L-amino acids, even in insects (See Kriel G. 1997. D-amino acids in animal peptides. Annu. Rev. Biochem. 66:337–45). The rare exceptions involve enzymatic post-translational modification. We have modified the sentence in the revised manuscript on lines 90-92 to include the qualifier ‘synthesized by ribosomes’ to remove this confusion. 

Rows 140-141. Please be more specific. In which sense do recording of clinical signs or behaviors of animals minimize suffering?

Response: We have added an explanation in lines 177-179 in the revised manuscript.

Row 153 In which case the tissue was not collected from animals. Please clarify this point.

In some animals only blood/plasma samples were collected, whereas in other animals tissue (skeletal muscle and brain) was collected. This has been clarified by the addition of text in lines 230-232 in the revised manuscript.

Row 159. Sample size is minimal acceptable thus statistical potency of the results low.

Response: Since repeated measurements along time were performed and only 3 data for each point a further analysis about statistical significance should be included, also authors should consider use of other tests such as repeated measurements ANOVA and post hoc tests.

Our primary outcome was changes in pharrmaokinetics determined by parameters such as Cmax, AUC and kel. Our sample size (n=3) was sufficient to yield statistical significance. This sample size is common in pharmacokinetic studies (Gabrielsson2016). We are interested in large, biologically, pharmacologically relevant effect; as we detected these with n=3, we are not concerned with smaller effects, which would be revealed by increasing sample size. It is useful to note that a test statistic is the product of sample size and effect size (Kühberger2015), in other words, p = f (ES, N), the effect size in these experiments was large enough to achieve a low p value (Kühberger2015). Therefore, if the effect size is small but the sample size very large, p will be small and therefore statistically significant. Conversely, if the effect size is large and the sample size small, p will also be small. For identical p values, given equal alpha levels, the probability of committing a Type I (false positive) error is the same regardless of sample size (Kühberger2015). That is, small sample requires a greater treatment effect than a large sample to obtain an equal level of statistical significance. Moreover, the mathematics of the significance test takes into account the sample size (Wilkinson1997). Indeed, Bakan (1966) stated that readers of research results need not adjust obtained significance levels according to sample size, for sample size has already been taken into consideration in the computation of the obtained significance levels. A larger sample size decreases the probability of committing a Type II error (false negative), which is not our concern given the effect sizes we have detected.

We did not use a repeated measures design, as 3 separate mice were used at each time point. Therefore, a repeated measures analysis is not appropriate. Moreover, ANOVA is not optimal as it doesn’t detect trends over time. Much preferred is to fit the data to a model, in this case a non-compartmental pharmacokinetics model (Gabrielsson2016). Doing so enables all points to contribute to the pharmacokinetic parameters. The parameters derived from the plasma vs time curve are what are conventionally analysed and statistically compared in pharmacokinetic analysis (Gabrielsson2016).

Please note that all our experiments have followed well-established and standard analysis for pharmacokinetic experiments and analysis (see Gabrielsson2016).

References:

Kühberger A, Fritz A, Lermer E, Scherndl T. 2015. The significance fallacy in inferential statistics. BMC Res Notes. 2015 Mar 17;8:84. doi: 10.1186/s13104-015-1020-4.

Gabrielsson J, Weiner D. Pharmacokinetic and Pharmacodynamic Data Analysis: Concepts and Applications. Fifth. Stockholm: Apotekarsocieteten; 2016.

Wilkerson M and Olson MR. 1997. Misconceptions about sample size, statistical significance, and treatment effect. The Journal of Psychology, 131:6, 627-631, DOI: 10.1080/00223989709603844

Bakan, D. 1966.The test of significance in psychological research. Psychological Bulletin, 66, 423-437.

More details about experimental design are needed since authors did not mention any positive or negative controls, also information about why they decided those timings for the samplings of the drug and also information about who they handled the tissues of the animal is needed.

Response: Again, please note that we have followed well-established and standard analysis for pharmacokinetic experiments and analysis (Gabrielsson2016). A negative control such as a placebo would be important for a pharmacodynamic experiment, but it is unclear how this would be appropriate or how it would be informative in a pharmacokinetic experiment. It is not clear what would constitute a positive or negative control in a pharmacokinetics study.

The timings were informed based on general parameters expected for oral absorption and elimination of typical small-molecule drugs. These timings are consistent with the pharmacokinetics of the racemate in humans, for which oral absorption results in detectable drug in the plasma at 15 min and is relatively slow followed by elimination with a half-life of 1-1.8 hours (Neuzil et al. 2002). Moreover, by virtue of the very data themselves, the timings are appropriate as values are obtained from which the desired pharmacokinetic parameters have been calculated, which was the objective of the study.

Authors refer to plasma samples but not tissue samples. So it is not clear at all who they measured and compared brain and muscle concentrations of the drug. Only in lines 265-269 some mention of this is done.

 Add information. 

Response: We have added a description of when the tissues were taken and processed in lines 230-232 in the revised manuscript.

Figure 4 is referred after figure 5 in the text. Please invert the order.

Response: We have inverted the order of the figures as requested in the revised manuscript.

Line 398, seems something is lacking "accounted for the by increase in Cmax"

Response: The wording was as we intended; however, as the reviewer’s response indicates that we were unclear, we have re-worded the sentence in the revised manuscript and hope the logic is now clear. The logic is that as AUC is the product of concentration and time, and when the rate of loss is the same (as in this case), the AUC is entirely dependent on where the decrease starts, which is dictated by Cmax.

Figure 3. The use of letters as symbols in the graphs is confusing and do not contribute to figure compression in fact it is cumbersome and did not allow comparison of the curves (same for figure 4).

Response: We disagree. The use of this type of graphic is to prevent the need for ‘decoding’ when looking at graphs. We adhere to and agree with the arguments against “encoded legends” and abide by the philosophy and practice that “every visual element contributes directly to understanding”. These principles of visual display are explained in detail by Tufte in ‘The Visual Display of Quantitative Information” and by John Tukey in "Exploratory Data Analysis".

figure 5 What is being compared in h, i and j.

Figure 5. What is being compared in h, i and j?

Response: The plotted values compared to the expected value. This is described in the legend for h and i. We missed stating the same for j, and have now added this text.

In figure 6 in Y Axes Use mg instead of ng

Response: Using mg would seem inappropriate. We assume the reviewer meant microgram? We thought deeply about what units to use, and although changing units would result in less clutter on some of the graphs (fewer zeros), one would have to decode and think about each set of numbers. This is similar to the arguments made for using abbreviation to save space, but in fact require more thought and interruption of the meaning (See: Tufte in ‘The Visual Display of Quantitative Information”). We decided to use ng/mL throughout the manuscript for complete internal consistency and to enable immediate and obvious comparisons between all the data in the paper.

Use of letters for symbols in the graphs seems not adequate it makes impossible to read value in brain for L and symbols are one in top of the other. Maybe use of bars would be adequate to rearrange this graphs and the whole figure.

Response: Again, we disagree and prefer to maintain the graphs as they are, to enable rapid and obvious comparison between the D and L, which is the primary objective. Changing the symbol from a letter to a square or circle does not solve the reviewer’s criticism, as the symbols would still overlap. Should a reader want to compare very small and accurate values, we have now provided all raw data in the Supporting Information.

Reviewer #2: The manuscript titled “Unexpected differences in the pharmacokinetics of N-acetyl DL-leucine enantiomers after oral dosing and their clinical relevance” by Strupp and colleagues details the pharmacokinetics of the racemate (N-acetyl-DL-leucine) or the nearly purified active L-enantiomer (N-acetyl-L-leucine) in mice. N-acetyl-leucine, as a racemic mixture, has been used as an over the counter anti-vertiginous drug and has also more recently been utilized in cerebellar ataxia, migraine, lysosomal storage disorders, and with age-related neural issues in the elderly. Recent evidence in animal models have indicated that the effectiveness of N-acetyl-DL-leucine is likely attributed to the L-enantiomer, N-acetyl-L-leucine. However, much is still to be worked out regarding the mechanism of action, pharmacodynamics, and particularly pharmacokinetics – all aspects of modern-day drug development that would further facilitate the use of N-acetyl-L-leucine or related analogs for treatment of the aforementioned medical issues, as well as possible improve their overall effectiveness. The data in this manuscript convincingly demonstrate that the pharmacologically active form N-acetyl-L-leucine has fundamentally different pharmacokinetics when given as a 50:50 racemate (N-acetyl-507 DL-leucine) when compared to administration of purified preparation containing mostly L-enantiomer and small amounts of the D-isomer. The pharmacokinetic results are consistent with probable inhibition of intestinal uptake of N-acetyl-L-leucine by N-acetyl-D-leucine in conjunction with selective first-pass metabolism of N-acetyl-L-leucine. Low levels of N-acetyl-L-leucine in the brain and muscle, even in the case of N-acetyl-L-leucine administration alone, suggest that N-acetyl-L-leucine may work as a prodrug that gives rise to the active agent upon metabolism. That metabolite is currently unidentified. The paper is reasonably well written and the data are statistically sound and clearly presented. Some statements need to be rephrased to improve clarity. I have listed only one moderate concern and several minor concerns below:

Moderate concern:

What is the rationale or motivation for using Balb/c mice? In light of recent concerns about sex as a biological factor, why were males mice only included in this study? How applicable are the pharmacokinetic data in mice to that in humans? Consider elaborating some of these details in the material and methods and/or discussion.

Response: These are very broad questions that are applicable to all pharmacokinetics studies in general and beyond the scope of our study. We have addressed these questions in the revised manuscript in a new paragraph (lines 202-210 in the revised manuscript by pointing the reader to the relevant literature and what is considered standard practice in the field.

Minor issues:

Abstract as it appears in the main body of the manuscript is incomplete in the abstract box on the title page. The first few sentences are missing. Not sure if this is was a post-uploading issue or inadvertent deletion from the authors.

Response: The same abstract is now present in both locations.

Abstract, pg 2, ln 49: Consider changing “…support the research and development of isolated N-acetyl-L-leucine” to “support the research and development of using only N-acetyl-L-leucine”.

Response: We have altered the wording as suggested in the revised manuscript.

Introduction, pg 3, Ln 55: Consider changing “as it is a” to “as a”.

Response: We have altered the wording as suggested in the revised manuscript.

Introduction, pg 3, Ln 56: Consider changing “cerebella” to “cerebellar”.

Response: We corrected the typo in the revised manuscript.

Introduction, pg 3, Ln 61: For clarity, consider rewriting “As N-acetyl-leucine is an analogue of the alpha amino acid leucine, it has a stereocentre and thus a pair of enantiomers” to “N-acetyl-leucine, an analogue of the alpha amino acid leucine, has a stereocentre and thus a pair of enantiomers”.

Response: We were trying to emphasize that the chirality of N-acetyl-leucine stems from the parent amino acid having a stereocentre and was not introduced by N-acetylation. We have reworded the sentence to clarify our meaning in the revised manuscript.

Introduction, pg 4, Ln 80: Consider changing “…from nice to know to effectively need

to know for informed dosing…” to “…from a “nice to know” to effectively “need

to know” basis for informed dosing…”

Response: We have altered the wording as suggested in the revised manuscript.

Introduction, pg 4, Ln 81-82: Consider simplifying “…it was and continues to be marketed as a racemate…” to ““…it was and continues to be used and marketed as a racemate…”

Response: We have altered the wording as suggested in the revised manuscript.

Introduction, pg 4, Ln 88-89: Consider inserting that and at in the following sentence: “each with distinct properties with one (N-acetyl-D-leucine) at best that does not contribute to the therapeutic response, and at worst is potentially responsible for toxicity, as this inactive enantiomer provides ‘therapeutic ballast’

Response: We have altered the wording as suggested in the revised manuscript.

Please elaborate on what you mean with the term “therapeutic ballast”. Is that synonymous with isomeric ballast?

Response: We have reworded this by adding the phrase ‘isomeric ballast’, which was coined by Ariens, and deleted the phrase ‘therapeutic ballast’ taken from the review by Lees. We also added a quotation from the Lees review that elaborates on the meaning. These changes appear on lines 112-118 in the revised manuscript.

Introduction, pg 4, Ln 96: Should “pharmacokinetic” be “pharmacokinetics”?

Response: Yes. We have corrected this typo in the revised manuscript.

MandM, pg 6, ln 141: Should “…distress, clinical signs and general behaviour of the

animals was recorded…” be “distress, clinical signs and general behaviour of the

141 animals were recorded…”

Response: We have corrected the verb tense in the revised manuscript.

MandM, pg 7, ln 153: “where no (ssue samples…” should be “where no tissue samples".

Response: We have corrected the typo in the revised manuscript.

MandM, pg 11, ln 248: Consider changing “quantitated” to “quantified”.

Response: We have changed the word in the revised manuscript.

Results, pg 13, ln 302: Consider rewriting “the main elimination was over for all enantiomers at around 2 h” as the elimination for the D-isomer is closer to four hours.

Response: We have changed 2 to 4 in the revised manuscript.

Results, pg 13, ln 326: “was the plasma is significantly higher” to “in the plasma is significantly higher”.

Response: We have corrected the typo in the revised manuscript.

Results, pg 15, ln 344: delete “compared” from “Another way to compare administration of the racemate compared to the purified enantiomer…” to read “Another way to compare administration of the racemate to the purified enantiomer…”

Response: We have corrected the typo in the revised manuscript.

Consider adding superscript asterisks to table to indicate level of significance on values where statistical tests were performed and are relevant.

Response: The table is provided as a reference for all the pharmacokinetic constants. As we made comparisons between two compounds in two formulations, there are six possible comparisons, of which we performed those that were scientifically sensible – that is, to reveal differences and interactions. It is very cumbersome to convey more than two comparisons in a table. Therefore, our statistical analyses are presented in Figure 5, where we can indicate the specific comparisons with lines and provide exact p values. 

6. PLOS authors have the option to publish the peer review history of their article (what does this mean?). If published, this will include your full peer review and any attached files.

Do you want your identity to be public for this peer review? For information about this choice, including consent withdrawal, please see our Privacy Policy.

Reviewer #1: No

Reviewer #2: No

---

## [Decision Letter · Decision Letter 1]

11 Feb 2020

Unexpected differences in the pharmacokinetics of N-acetyl DL-leucine enantiomers after oral dosing and their clinical relevance

PONE-D-19-24169R1

Dear Dr. Churchill,

We are pleased to inform you that your manuscript has been judged scientifically suitable for publication and will be formally accepted for publication once it complies with all outstanding technical requirements.

With kind regards,

Nicolás Pérez-Fernández

Academic Editor

PLOS ONE

Additional Editor Comments (optional):

Reviewers' comments:

Reviewer's Responses to Questions

**Comments to the Author**

1. If the authors have adequately addressed your comments raised in a previous round of review and you feel that this manuscript is now acceptable for publication, you may indicate that here to bypass the “Comments to the Author” section, enter your conflict of interest statement in the “Confidential to Editor” section, and submit your "Accept" recommendation.

Reviewer #1: All comments have been addressed

Reviewer #2: (No Response)

2. Is the manuscript technically sound, and do the data support the conclusions?

Reviewer #1: Yes

Reviewer #2: Yes

3. Has the statistical analysis been performed appropriately and rigorously? 

Reviewer #1: I Don't Know

Reviewer #2: Yes

4. Have the authors made all data underlying the findings in their manuscript fully available?

Reviewer #1: Yes

Reviewer #2: Yes

5. Is the manuscript presented in an intelligible fashion and written in standard English?

Reviewer #1: Yes

Reviewer #2: Yes

6. Review Comments to the Author

Reviewer #1: I suggested a change in the figure presentation but authors decided not to follow that recommendation.

I recommend the authors to review the work from:

Calin-Jageman RJ, Cumming G. Estimation for Better Inference in Neuroscience. eNeuro. 2019 Aug 1;6(4). pii: ENEURO.0205-19.2019.

Reviewer #2: I thank the authors for addressing most of my concerns. I only have a few minor suggestions:

Pg. 4, Ln 90: insert "a" before "nice to know"; hyphenate "nice-to-know" and "need-to-know".

Pg. 7, Ln 154: "Noninvasively" should be "noninvasive".

Pg. 8, Ln 174: Consider changing "because this inbred strain of mice as it commonly used" to read "...because this inbred strain of mice is commonly used..."

Figure Legend 3: Consider stating that "Plotted values at each time point are the mean +/- SEM from three mice". Repeat similar phrasing in other figure legends. Otherwise it suggest that only three mice were used for the total plot.

Pg. 19, Ln 434: Consider inserting "likely" before "interfering".

Pg. 20, Ln 460: Change the "a" after "capacity" to a comma.

Pg. 20, Ln 462, "acids are taken up...." should be "acids taken up..."

Pg. 20, Ln 480: Should "acylase" be "acetylase"?

7. PLOS authors have the option to publish the peer review history of their article (what does this mean?). If published, this will include your full peer review and any attached files.

Reviewer #1: No

Reviewer #2: No

---

## [Editor Report · Acceptance letter]

18 Feb 2020

PONE-D-19-24169R1 

Unexpected differences in the pharmacokinetics of N-acetyl-DL-leucine enantiomers after oral dosing and their clinical relevance 

Dear Dr. Churchill:

I am pleased to inform you that your manuscript has been deemed suitable for publication in PLOS ONE. Congratulations! Your manuscript is now with our production department. 

With kind regards,

on behalf of

Dr. Nicolás Pérez-Fernández 

Academic Editor

PLOS ONE